# Classification of Low Earth Orbit (LEO) Resident Space Objects’ (RSO) Light Curves Using a Support Vector Machine (SVM) and Long Short-Term Memory (LSTM)

**DOI:** 10.3390/s23146539

**Published:** 2023-07-20

**Authors:** Randa Qashoa, Regina Lee

**Affiliations:** Department of Earth and Space Science, York University, Toronto, ON M3J 1P3, Canada; reginal@yorku.ca

**Keywords:** resident space object, Space Situational Awareness, light curve, low Earth orbit, support vector machine, long short-term memory

## Abstract

Light curves are plots of brightness measured over time. In the field of Space Situational Awareness (SSA), light curves of Resident Space Objects (RSOs) can be utilized to infer information about an RSO such as the type of object, its attitude, and its shape. Light curves of RSOs in geostationary orbit (GEO) have been a main research focus for many years due to the availability of long time series data spanning hours. Given that a large portion of RSOs are in low Earth orbit (LEO), it is of great importance to study trends in LEO light curves as well. The challenge with LEO light curves is that they tend to be short, typically no longer than a few minutes, which makes them difficult to analyze with typical time series techniques. This study presents a novel approach to observational LEO light curve classification. We extract features from light curves using a wavelet scattering transformation which is used as an input for a machine learning classifier. We performed light curve classification using both a conventional machine learning approach, namely a support vector machine (SVM), and a deep learning technique, long short-term memory (LSTM), to compare the results. LSTM outperforms SVM for LEO light curve classification with a 92% accuracy. This proves the viability of RSO classification by object type and spin rate from real LEO light curves.

## 1. Introduction

This section provides background information on light curves, the dataset we are using, and the methods we are applying to obtain features from light curves in order to classify them.

### 1.1. RSO Light Curves

Light curves represent the brightness of objects (stars, RSOs and planets) as a function of time. They are commonly used in various space applications (astronomy, planetary studies, etc.) to understand the characteristics of celestial objects. Various studies on SSA, such as [1,2,3,4,5], rely on light curve information to extract the properties of RSOs in the trends of this time series data; as such, extracting RSO features from these data are proven to be very valuable. Light curves have also been used for estimating ballistic coefficients for LEO RSOs, as proven in [6]. Additionally, light curves can be simulated as was performed by [7,8,9,10], which allows researchers to obtain large quantities of data on which to perform analyses.

Assessing an RSO’s stability (spin rate or pointing stability) is useful as we identify the characteristics of RSOs through light curves even when we don’t know the identity of the RSO we are observing. Once the stability of the RSO has been determined, we classify the RSO into various categories. Figure 1 illustrates the commonly accepted RSO classes.

There have been multiple studies on the use of ontology for RSO characterization and classification using the classes listed in Figure 1 [11,12,13]. The most basic classification is by object type. RSOs include satellites, rocket bodies, or other debris. Satellites are further classified by the presence or absence of an active control system which generates two subcategories: stable and tumbling satellites. Another common label for RSOs is the orbit type, which is determined from the altitude of the orbit. There are three main orbit types, LEO, medium Earth orbit (MEO), and high Earth orbit (HEO). A very common HEO orbit is a geostationary orbit (GEO), which is typically placed as a separate category. GEO orbits are unique as they have an altitude of 35,786 km and an orbital period that equals Earth’s rotation around itself. RSOs are also classified by their geometry and the direction they are pointing at. When considering observations of space objects, there are more categories that are useful in SSA, the most important of which is the material an RSO is made of, which impacts its reflection patterns. Another important parameter is the orientation, as the shape of the RSO as perceived by the observer appears different depending on its attitude. This also affects studies of brightness, as the reflection from a solar panel is different from that of the bus.

### 1.2. Study Dataset

The dataset used in this research consists of light curves collected from the Ukrainian database and atlas of light curves [14]. This database contains a collection of light curves from 340 (LEO) RSOs with observations performed from 2012 to 2020. Satellites represent 72.9% of the RSOs in the database, where 30.3% of all the RSOs are stable satellites, 18.5% are tumbling satellites, and 24.1% have an unknown status at the time of observation. Other RSOs in the database consist of rocket bodies which represent 20.3% of the total RSOs and debris which are the remaining 6.8%. Since the percentage of debris is much lower than the other categories, debris light curves are not included in this study. The light curves were collected from a KT-50 telescope in Odessa [14]. For each RSO, there are 1–343 observations taken at different time periods across a 9-year time span. Most light curves have observation lengths of less than 15 min. The light curves are in the form of a matrix containing the modified Julian date and brightness for a specified North American Aerospace Defense Command (NORAD) catalog number that identifies the studied RSO. For visualization purposes, the light curves for each RSO were plotted. Examples of light curves from RADARSAT-2 on 30 November 2016, as a sample stable satellite, Jason-2 (OSTM) on 11 September 2020, showcasing a tumbling satellite, and CZ-4C on 14 July 2020, which is a rocket body, are depicted in Figure 2, Figure 3 and Figure 4.

Both RADARSAT-2 and Jason-2 (OSTM) are boxwing satellites, while CZ-4C is expected to be a typical rocket body with a cylindrical body and a conical tip. Figure 2 has three distinct peaks which we assume to be caused by the antenna and solar panels. Figure 3 has many peaks, which indicates that the RSO completes more than one rotation during the observation. This is typical for tumbling satellites as they rotate much more quickly than stable satellites.

As for Figure 4, it shows a very different pattern where there are no distinct or wide peaks and the brightness constantly drops. By observing the light curve more closely, it appears that there are multiple local peaks despite the overall brightness decrease during the observation. This behavior is common among all rocket body light curves. The similarity in the brightness pattern could be attributed to the fact that the shape of the RSO is similar on multiple sides and there is no face that is significantly more reflective, as opposed to solar panels and antennas on satellites.

Aside from attitude and shape, phase angle, which is the sun–RSO–observer angle, is typically referenced when analyzing light curves. For tumbling RSOs such as tumbling satellites and rocket bodies, the phase angle continuously changes as we are looking at different faces continuously. However, for stable satellites, we do not expect the phase angle to change as much. This applies to the light curves in Figure 2, Figure 3 and Figure 4 as there is only a 10${}^{\circ }$ variation in the phase angle during the observation in Figure 2, but that number increases significantly to about 80${}^{\circ }$ for Figure 3 and Figure 4.

It should be noted that these assumptions are applicable to LEO RSOs only, as GEO RSO light curves have not been analyzed in this study. Overall, we can assess that LEO light curves of stable satellites have very few peaks that are attributed to the surfaces with highest reflectivity. Tumbling satellite light curves have multiple distinct peaks or glints that have a repeated pattern. As for rocket bodies, their light curves have an almost linear brightness pattern with no clear peaks. Additionally, stable satellites have relatively small changes in phase angle, as opposed to tumbling satellites and rocket bodies.

### 1.3. Literature Review on RSO Light Curve Classification

Many approaches are used for light curve classification, with machine learning being one of the most advanced and popular methods. The study in [15] provides a summary of different techniques of RSO classification. When researching different machine learning techniques for RSO light curve classification, there were two main types of studies. The first involves classifying light curves into specific known RSOs, these could be specific satellites, known satellite buses, or even certain rocket bodies. This type of research is demonstrated in [16,17]. Dual-band simulated LEO light curves were classified in [16], which presents a unique approach in the literature for light curve classification. The main limitation with this form of classification is that it is used to identify a limited number of RSOs, and it is difficult to extend to the full NORAD RSO catalog as there are over 50,000 RSOs in the catalog.

The other form of light curve classification characterizes a variety of RSOs into different sets of categories such as shape, object type, or stability, to indicate if an RSO is actively controlled or not. This is the form of classification applied in this research. There are three aspects of this study that, when combined together, highlight the novelty of this work. These three points are: (1) LEO light curve analysis, (2) real light curve analysis, and (3) wavelet light curve analysis, which will be described in this section.

#### 1.3.1. LEO Light Curve Analysis

Most studies on light curve classification are performed on objects in GEO due to the wide availability of measured light curves in addition to the availability of long duration observations of several hours. The works in [18,19,20,21] serve as examples of that. The work in [19] first attempts classification on a dataset of GEO light curves before applying the same algorithm on real light curves from a variety of orbits, including LEO and GEO. Studies on GEO light curves serve as a baseline for the design of LEO light curve classification algorithms, although the implemented strategies may not be the same due to the differences in the relative brightness, size, and motion of RSOs.

For these reasons, LEO RSOs have not been studied as extensively as GEO light curves but, with the database and methodology used in this research, much more detailed analysis can be performed. The biggest difference between the dataset in this research and the ones used in GEO classification papers is that most observation durations are short, typically no more than a few minutes, with the longest observation being around 15 min in this study. There are only a handful of studies that discuss the classification of LEO light curves using neural networks, such as [22]. In [22], extra steps were taken for classification, including applying a test for periodicity and aliasing, aside from extracting the needed features using wavelet decomposition. In this work, we do not need to apply these extra steps and use the features from the wavelet scattering transformation to train a machine learning algorithm with comparable accuracy. In [22], RSO characterization is performed using shape and spin rate of simulated light curves. However, in this study, we classify RSOs by object type and spin rate. The classifier sorts RSOs as stable satellites, tumbling satellites, and rocket bodies in this study, whereas the study in [22] classifies objects into stable objects, tumbling objects, or tumbling fragments. A tumbling object includes both satellites and rocket bodies and does not distinguish between the two.

#### 1.3.2. Real Light Curve Analysis

There are many studies on light curve simulations, such as those in [7,8,9,10]. Several light curve classifiers are trained on simulated data with the expectation that it would closely match real light curves but, as indicated in [19], this is hardly the case. In [19], a convolutional neural network (CNN) classifier initially designed for simulated light curves was trained on real light curves, which resulted in a 75.4% accuracy on real light curves and 97.83% on simulated light curves. Therefore, to build a classification algorithm that performs well on light curves collected in the field, real light curves must be used to train the algorithm. Although the study in [22] was on LEO light curves, analysis was only performed on simulated light curves in [22], whereas we use observational light curves.

#### 1.3.3. Wavelet Light Curve Analysis

In order to extract the characteristics and features from light curves, we use wavelet transformation. Wavelet transformation is a frequency analysis method that is comparable to fast Fourier transformation (FFT). As opposed to FFT, which decomposes a signal into a collection of sinusoids, wavelet transformation breaks down the original signal into multiple wavelets, which are signals that are scaled and shifted with regards to a known signal or mother wavelet [23]. These wavelets are finite, as opposed to sinusoids that continue indefinitely. There are two main issues with the Fourier analysis listed in [24]: spectral leakage and Gibb’s phenomenon. Spectral leakage is an issue where noise occurs during frequency analysis due to zero padding or other non-frequency related changes to the signal. This occurs because the finite inputs are then converted during Fourier analysis to a set of infinite sinusoids [24]. Gibb’s phenomenon occurs when trying to fit a signal with abrupt changes and steep slopes to a set of sinusoids which are essentially smooth signals [25]. As a result, FFT does not work well for discontinuous signals and those that do not have enough samples for analysis. When analyzing observational light curves, wavelet analysis is used to address these shortcomings. In [22], a discrete wavelet transformation (DWT) was used to obtain features from a light curve following the periodicity and aliasing test. In this study, we propose a different method of using wavelets.

To obtain a set of features from the light curves that are used for training machine learning algorithms, a technique called wavelet scattering transformation (WST) is used. WST is a technique that combines iterative convolution with wavelet transformation and low-pass filtering [26]. To apply WST, the signal is first convolved with the scaled mother wavelet. Then, the modulus of the convolved signal is applied to recover the information lost due to down sampling. Lastly, a low pass filter is applied to the resulting signal [26]. Using WST has multiple advantages, including stability in the face of signal time-warping deformations and noise, thus making it ideal for classification [27]. There have been multiple studies on applying WST to obtain features to classify ECG signals, which also consist of time series data, including [28,29,30]. In this study, we apply a similar technique to obtain features for LEO RSO light curve classification. The key contribution from this study is that measured LEO light curves are classified into three categories using only one major step for pre-processing: wavelet scattering transformation. This research tests the application of two different machine learning algorithms for classifying observational LEO light curves.

### 1.4. AI Implementation of Time Series Classification

In this study, two different artificial intelligence (AI) algorithms are applied for LEO light curve time series classification to determine the type of RSO. We first start by applying an SVM. Then, we perform the same classification with a neural network, specifically an LSTM algorithm. By studying two different types of AI algorithms, we determine the method that is best suited for light curve classification of RSOs.

#### 1.4.1. SVM

SVM is commonly used for classification and regression for multiple reasons. Unlike neural networks, SVM is computationally efficient. SVM creates a hyperplane, essentially a multi-dimensional straight line, that separates the different classes. The defining feature of an SVM is the approach of selecting the hyperplane [31]. SVM adopts a concept called the maximum-margin hyperplane, where it selects the hyperplane that maximizes the distance between the different classes such that it is in the middle. When the trends in the data are multi-dimensional, a kernel is used which adds extra dimensions as needed for improved performance. SVM is also applied for multi-class classification where there are more than two output cases. An in-depth analysis of SVMs is available in [32]. An SVM has been used in astronomical research to classify light curves of stars as in [33], so it should also be sufficient for LEO RSO light curve classification.

#### 1.4.2. LSTM

LSTM is a type of recurrent neural network (RNN) that resolves the vanishing and exploding gradient issue typically associated with RNN [34]. LSTM consists of three gates or parts. The first determines the signal at the current step, the second determines the signal at the previous step, and the last is the “forget” gate which determines how much of the previous signal to consider in the current signal. Combining previous and current steps makes LSTM a robust tool for analyzing time series data [35].

LSTM has been widely used for time series classification, such as in [36,37]. It has also been applied to similar research problems, such as star classification [38]. For LEO RSO light curve classification, LSTM has been used in combination with a hidden Markov model (HMM) in [22] for simulated light curve classification. To train an LSTM, there are multiple hyperparameters that must be carefully selected to obtain the best performance. There are a wide range of hyperparameter values to choose from, which makes the task even more daunting. There are multiple methods to choose the best hyperparameter values, but the method implemented in this study is Bayesian optimization.

### 1.5. Bayesian Optimization

Bayesian optimization is applied to obtain the hyperparameter values that result in the best algorithm performance and is especially useful when the function we need to minimize is overly complex as it considers the function as a black box. The goal of Bayesian optimization is to find the set of requested parameters that minimizes the value of an unknown objective function [39]. Bayesian optimization is considered a Gaussian process where the value of the next iteration is based on the current iteration [40]. Bayesian optimization is also much less computationally expensive than an exhaustive sweep over all potential options. A more extensive analysis of using Bayesian optimization for choosing hyperparameters is in [41,42].

## 2. Generating Training Features from Light Curves

This section explains the steps taken to obtain the training features from the light curves used in this study.

### 2.1. Data Formatting and Extraction

Prior to applying WST, the data from [14] required some formatting to be more intuitive. In the original data, there is one file for each RSO with numerous observations taken at different dates listed sequentially. For this research, we separated each observation into a separate input file. This makes plotting and subsequent analysis much more intuitive as different RSOs are detected at different times.

Aside from the multiple observations per file, there were two main issues in the original data. Each observation is typically sorted in order of increasing time but there are some that stray from this norm. There are duplicate measurements that are unsorted. The lack of sorting is problematic when performing any form of time series analysis, so the pre-processing code contains an extra feature to sort the data in each observation. Another, more serious, concern is that there are some observations with a constant magnitude of 36. This is a critical issue as it greatly skews the results of subsequent analyses. This was remedied by first locating and then removing the anomalous observations during pre-processing. The light curves described in this study have all been formatted by this method. The next step involves extracting the labels. The dataset contains light curves accompanied by the NORAD catalog number which indicates the RSO’s identity. To store the needed labels for a large collection of RSOs, we developed a script that would obtain the needed information from several websites, such as Gunter’s Space Page and N2YO, and store the label in an Excel sheet.

Given the large amount of data needed for this study, typical methods of loading variables into memory are very computationally expensive, so we developed a custom datastore that records the location of each light curve file alongside its label. The label contains the class of the RSO, which is one of three options: stable satellite, tumbling satellite, or rocket body. This allows us to perform the needed transformations on the data while out of memory for most of the code and only reading the data into memory at the last step when obtaining the features. More information on datastores is found in [43,44].

### 2.2. Feature Extraction

As mentioned in Section 1.3.3, wavelet scattering transformation is applied to the light curves to generate a set of training features to be used for classification. In order to create the scattering network, the characteristics of the inputs need to be provided. This includes specifying the signal length, invariance scale, and sampling frequency. The signal length represents the number of samples available. The invariance scale refers to the limit of the low pass filter beyond which the signal is reduced to zero. The sampling frequency is the rate of capturing data, which has been determined as approximately 48 Hz for the dataset used in this study.

After creating the scattering network, the scattering feature matrix is obtained. To generate valid inputs for classification, a constant signal length must be chosen. Given that the light curves have varying durations, a single signal length must be chosen, and the remaining data must be padded or truncated. There are many ways to choose the signal length. In this study, the value that maximizes the training accuracy while having enough training inputs was chosen. If a light curve has been collected for longer than the specified duration, it is truncated. As for shorter duration light curves, there are two possible ways for them to be pre-processed. The first method simply eliminates all light curves with durations shorter than the specified duration. Alternatively, the shorter signal is padded with zeros to have the applicable number of samples. For this study, the shorter duration light curves have been removed but we have included the option for padding the signal in the algorithm if it is deemed necessary for future studies. Additionally, if required by the dataset, normalization is applied but, for this study, it made no impact due to the consistency of the inputs. The features consist of a 3D matrix with the first dimension indicating the number of scattering paths, the second referring to the number of scattering coefficients, and the third dimension referring to the total number of light curves.

### 2.3. Input Data Visualization

To better visualize the different classes of light curves, we implemented a t-distributed stochastic neighbor embedding (t-SNE), which is a statistical technique for converting multi-dimensional data into two or three dimensions, which is easier to visualize [45]. Similarly structured data appear to be clustered together, which differentiates between the different object classes. Unlike principal component analysis (PCA), which is a linear technique, t-SNE is a non-linear technique that preserves the local trends in the data. There are multiple distance metrics that can be used to compute the t-SNE, but in this study we implemented the Euclidean and Chebychev distance equations on the light curves of the three different classes to generate a 3D embedding of the input data shown in Figure 5 and Figure 6.

Figure 5 and Figure 6 reveal different clusters in the data. For example, most stable satellites are clustered together in the Euclidean embedding. This does not hold for the Chebychev distance calculation though, where stable satellites and rocket bodies appear to cluster in a similar region. By applying t-SNE, we can see some separation among the different classes but it is not very pronounced. This makes it difficult for classification. After applying WST to match the input to the different algorithms, the relationship between the different coefficients now has a different trend, which is shown in Figure 7 and Figure 8.

After applying t-SNE on the WST features, we can notice a closer similarity between the Euclidean and Chebychev distance approaches. This is purely an artifact of the data though, and does not necessarily indicate an underlying trend in the data. The classes are still not completely split into individual clusters, but we can notice that certain cluster regions now appear for the three different classes. Given that the classes are not appearing in tightly grouped clusters, the light curves will not be simple to classify and there may be some outliers in the classification results.

## 3. Light Curve Classification

To classify light curves, we first implemented an SVM with two different methods of preparing the inputs to study the difference in performance. Then, we trained an LSTM RNN to assess the difference between a simple machine learning approach and a neural network. To compare the results, we computed some performance metrics such as accuracy, precision, recall, and F1 score. Since we have a multi-class classification problem with a different number of elements per class, weighted averaging was used. The definition and equations for each of these terms are defined in detail in [46,47].

### 3.1. SVM Results

When implementing an SVM, we utilized two different approaches for generating the model inputs. We attempted a mean features approach and a window-based approach. For both types of inputs, we defined the same training parameters. To obtain higher accuracy results, we implemented a second order polynomial kernel as opposed to using a linear function. The next factor to choose was the type of multi-class classifier. When there are more than two output classes, there are two main ways to specify how the computation happens. The first approach is a one vs. all classification, where each class is compared with all other classes [48]. The second approach, called one vs. one, breaks the multi-class classification problem into a collection of binary classifications, where each class is compared to one class at a time forming pairs. A deeper comparison of the two techniques is in [49]. For this study, we showcase the results of using a one vs. one classification approach as we noticed minimal change on the testing accuracy from a one vs. all approach.

#### 3.1.1. Window Based Approach

The first method of preparing inputs for SVM we attempted is by using a window-based approach. To perform this, we split each scattering time window as a separate input to the model and trained the SVM. When computing the performance metrics, we combined the signals back to the initial input size and took the label featured most for each RSO as the predicted label. The overall accuracy of this method was 60%, which is unacceptable for classification. Table 1 summarizes the performance metrics. The performance of the algorithm is poor specifically when considering precision, recall, specificity, and F1 score.

#### 3.1.2. Mean Features

The second type of feature pre-processing involves using the mean features from WST as inputs to the SVM. This allows us to obtain a 2D set of features from a 3D array. The average of the columns, which consist of the second dimension of the matrix, is taken such that an m by n by o sized-matrix becomes an m by o sized array. By reducing the dimensional of the input, we reduce computation time significantly. The resulting test accuracy is 87%.

The overall performance of the classifier when using mean features has improved significantly as compared to the window-based approach (see Table 1), and the algorithm runs about 86% quicker as it only takes 7 s to train the network. Given that the values of all metrics are relatively similar to each other, we conclude that this approach results in more consistent classification regarding false positives and false negatives.

### 3.2. LSTM Implementation

To study whether a neural network performs better LEO light curve classification as compared to a simpler approach like SVM, we trained an LSTM. This method was chosen specifically due to its wide use in the literature for multi-class classification. As opposed to SVM, which requires a few options to choose from before training, neural networks require the user to supply many hyperparameters, and the sheer number of potential options require some form of optimization technique to determine the best values to choose from. As a result, we implemented Bayesian optimization to select the best values for the hyperparameters.

#### 3.2.1. Applying Bayesian Optimization

There are many hyperparameters that we could optimize but, for this analysis, we chose to prioritize the number of hidden units, the maximum number of epochs, and the mini batch size, as well as the initial learning rate. The number of hidden or recurrent units refers to the number of internal layers within the LSTM network. As indicated in [50], if this number is too large, the network overfits the data quickly, which deteriorates the performance on a test set, but if the number is too small, then the network does not store enough details to efficiently solve the problem. The number of epochs refers to the number of times the model passes through the complete training set. Similar to the number of hidden units, too small a value misses essential features of the data, while too large a value risks overfitting [51]. Mini batch size refers to the number of inputs to use for each iteration. Thus, each epoch has a collection of batches. Finally, the learning rate indicates the change in weight of the model. A very small value takes too long to train and a very large value risks overshooting and diverging from the expected result.

#### 3.2.2. LSTM Results

After obtaining the hyperparameter values using Bayesian optimization, which took over a week to optimize 30 epochs, the LSTM network was trained, which resulted in 92% accuracy. Additional performance metrics are provided in Table 1. Training the LSTM was completed in 3.5 min, which is 30 times longer than SVM mean features. Although this algorithm is slower than the mean features SVM approach, LSTM has the best performance as compared to both SVM approaches with regards to all five performance metrics. This means that, although simpler solutions like SVM work relatively well for light curve classification, higher accuracy demands more complex methods such as deep learning with LSTM used as an example in this study.

## 4. Conclusions

In this study, we implemented a novel approach for LEO RSO classification of observational light curves. We used wavelet scattering transformation to extract features from LEO light curves, which was then applied to different AI algorithms to classify RSOS by object type (stable satellite, tumbling satellite, and rocket body). By visually observing the different light curves, we determined some differences between stable satellites, tumbling satellites, and rocket bodies, but these differences are complex, which make it difficult to apply analytical classification. This is further proven by applying t-SNE to the input light curves as well as the WST coefficients, as the clusters are closely grouped together. Applying t-SNE on the WST coefficients adds more distinction between the classes, but the result is not pronounced enough to warrant simpler classification techniques feasible. We applied a non-deep learning technique (SVM) as well as a deep-learning approach (LSTM) to compare classification performance.

We used the scattering network features to train both the SVM and LSTM for comparison. For the SVM, we first tested a window-based approach where the scattering network was split by each individual time window. Although we expected this method to be satisfactory for classification, the resulting accuracy was only 60%. The low accuracy is most likely due to the lack of sufficient features in each time window and the resulting large amount of training data due to splitting each time window as a separate input. SVM does not work well for very large datasets, and it seems to be the main cause of the poor performance. The second approach to SVM used the mean features of the scattering network as input parameters. Although we removed one dimension from the data which decreased computation time and simplified the inputs, the accuracy has increased significantly to 87%. This also proves that the large dataset was the reason for the time window approach failing. Finally, we trained an LSTM on the features from the scattering network as is. A major complexity of neural networks is determining the ideal value of the hyperparameters that result in the best performance and, in this case, Bayesian optimization was used. Using the hyperparameters from Bayesian optimization resulted in the highest accuracy out of all approaches, 92%. This is significant to the SSA community as this study provides the capability of classifying observational LEO light curves into stable satellites, tumbling satellites, and rocket bodies.

Future work includes implementing (1) a light curve classification algorithm on datasets of observational light curves taken from a space-borne sensor, as well as looking at (2) the improvement of classification accuracy for each class. For space-borne datasets, we are currently examining Fast Auroral Imager (FAI) and Near Earth Object Surveillance Satellite (NEOSSat) images. FAI is an instrument onboard the CASSIOPE satellite [52]. As indicated by its name, the original purpose of FAI was to study auroras, but the camera specifications work for RSO observation as well. The camera specifications are comparable to a star tracker in that it has a low resolution of 256x256. NEOSSat is a Canadian satellite that targets asteroids [53]. Ground-based images are captured from a fixed observer, but space-based images are captured from a moving, and sometimes spinning, satellite. Therefore, space-based images are much more complicated for light curve extraction and classification. Another aspect that impacts the generation of light curves is the method of capturing images. An observer can either point at a fixed location and capture images, which we refer to as stare mode, or track RSOs as they pass through the field of view, referred to as tracking mode. Light curves have traditionally been captured in tracking mode, but we are interested in attempting stare mode light curve extraction due to the wide availability of such images.

A lack of labeled data continues to be the main challenge in improving classification accuracy. As in any other AI-related studies, properly annotated and accurately labeled datasets are critical to improve algorithm accuracy. We are currently improving a star-field simulator with RSO light curve information to provide extensive, accurate, and high fidelity simulated image sets. In parallel, we have also initiated the effort to collect light curve data using our own image sensors to prepare datasets. We have collected a multitude of ground-based images from field campaigns as well as near-space images from a stratospheric balloon mission, and we are currently looking into the best method to extract light curves from each dataset.

There are other methods available for improved classification accuracy. Studies into building a two-tier classifier could be performed where the first level classifies an RSO into a satellite or a rocket body and the second tier classifies the satellite into stable or tumbling, which might result in improved performance. Transfer learning is another avenue of future research which might improve the results of implementing the developed algorithm on other datasets. Lastly, we also consider adding debris as a fourth category for the classification.

## Figures and Tables

**Figure 1 sensors-23-06539-f001:**
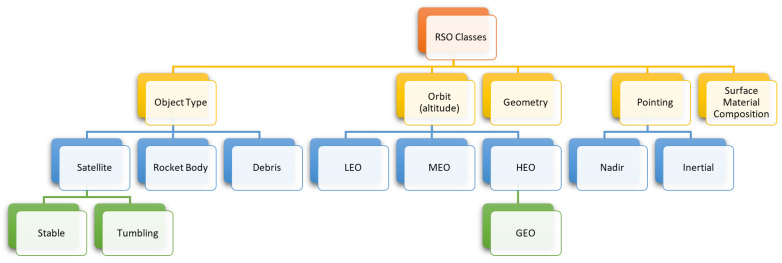
RSO classes.

**Figure 2 sensors-23-06539-f002:**
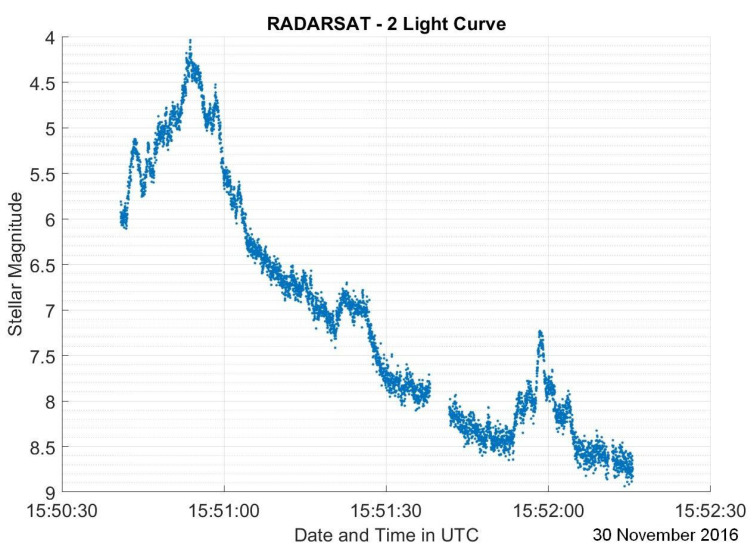
RADARSAT-2 light curve on 30 November 2016.

**Figure 3 sensors-23-06539-f003:**
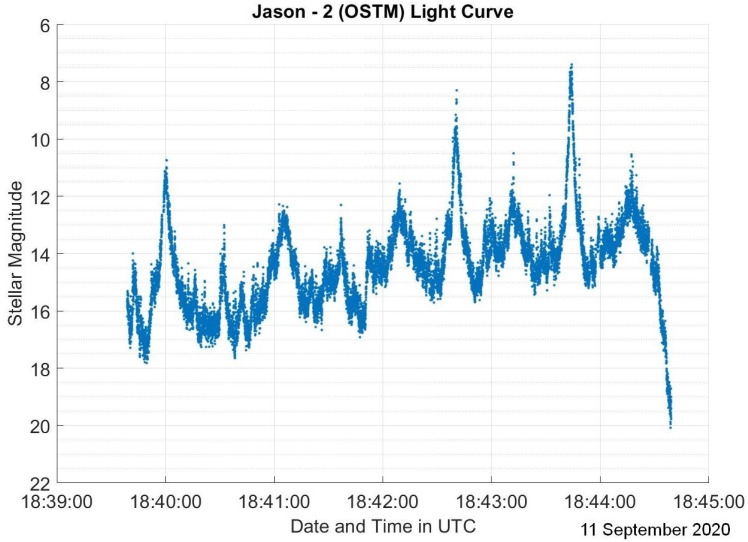
Jason-2 (OSTM) light curve on 11 September 2020.

**Figure 4 sensors-23-06539-f004:**
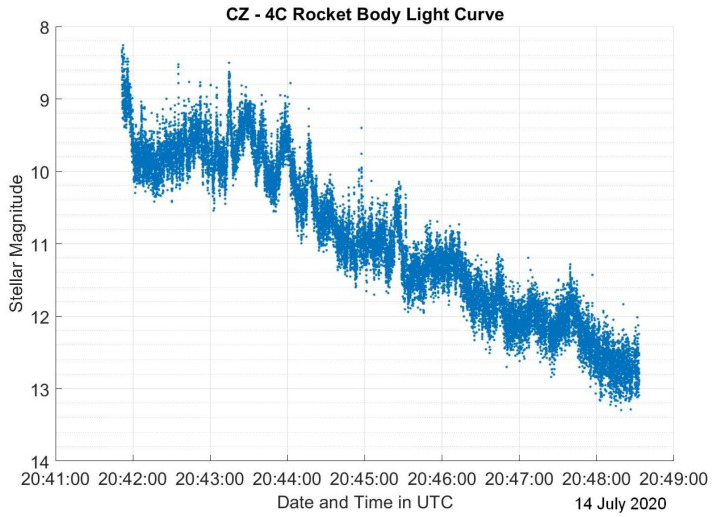
CZ-4C light curve on 14 July 2020.

**Figure 5 sensors-23-06539-f005:**
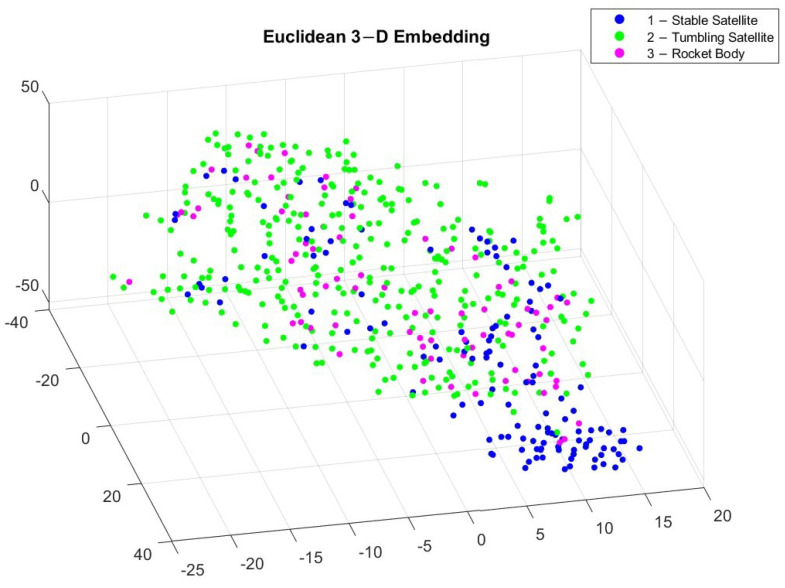
3D embedding of light curves using Euclidean distance.

**Figure 6 sensors-23-06539-f006:**
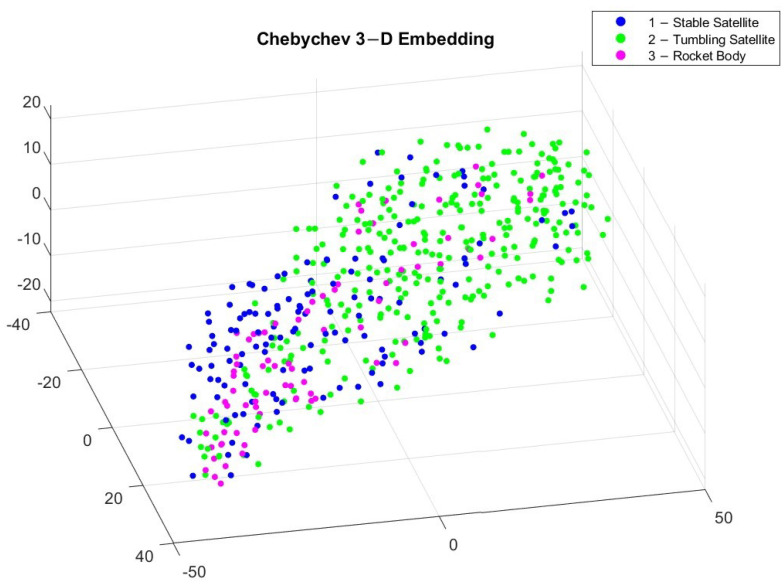
3D embedding of light curves using Chebychev distance.

**Figure 7 sensors-23-06539-f007:**
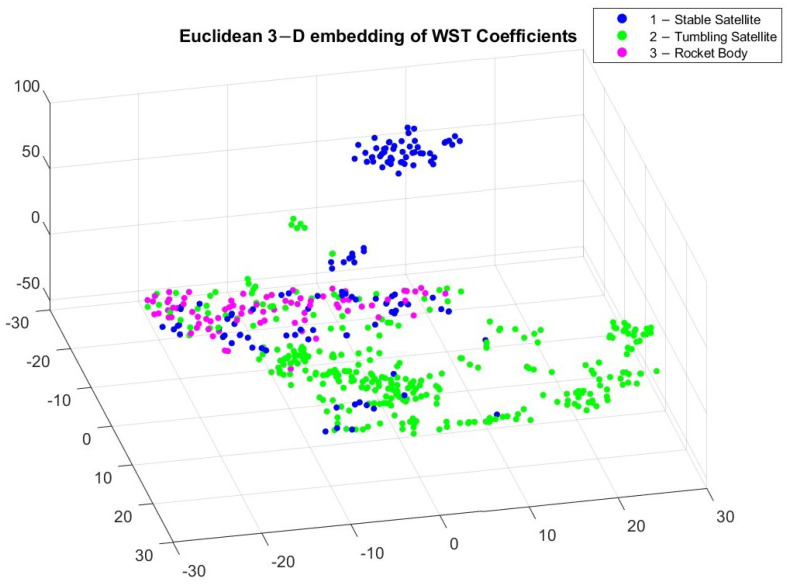
3D embedding of WST coefficients using Euclidean distance.

**Figure 8 sensors-23-06539-f008:**
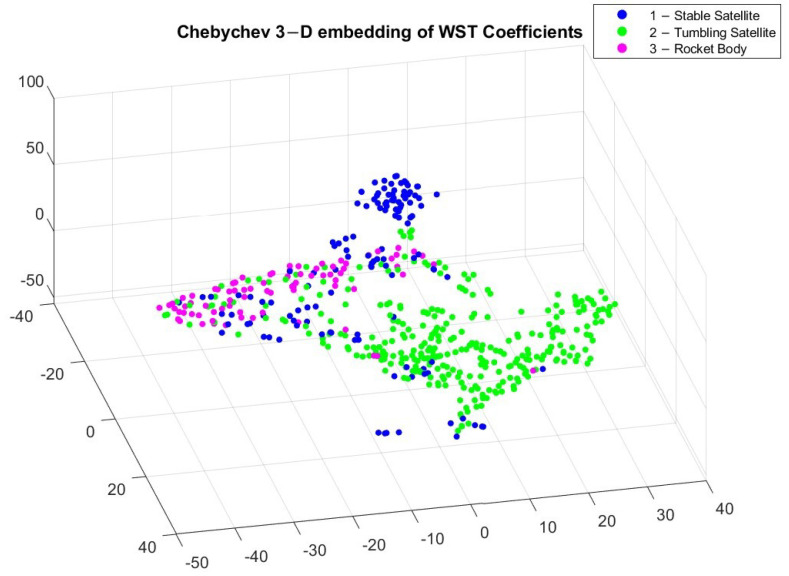
3D embedding of WST coefficients using Chebychev distance.

**Table 1 sensors-23-06539-t001:** Performance metrics of different light curve classifiers.

Method	Accuracy	Precision	Recall	Specificity	F1 Score
SVM Window-Based Approach	60%	40%	48%	46%	40%
SVM Mean Features	87%	86%	82%	84%	81%
LSTM	92%	90%	89%	95%	89%

## Data Availability

Restrictions apply to the availability of these data. Data was obtained from University of Texas at Austin and are available from Prof. Moriba Jah with the permission of University of Texas at Austin.

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
