# Peer review of "Classification of Low Earth Orbit (LEO) Resident Space Objects’ (RSO) Light Curves Using a Support Vector Machine (SVM) and Long Short-Term Memory (LSTM)"

_sensors, 2023, doi:10.3390/s23146539_

Round 1

Reviewer 1 Report

This study carries out lightcurve classification of LEO objects using SVM and LSTM methods.

Section 1 contains the introductory material on RSO classification, the current dataset, 

review of RSO lightcurve classification and the two applied AI methods.

Section 2 discusses feature extraction for classification (preprocessing the data).

Section 3 presents the classification.

Section 4 gives the conclusions.

 line 138: "As a result, FFT does not work well for signals that continuously change and those that do not have enough samples for analysis."

On the contrary FFT works well for continous signals but not on discontinuous signals. Please correct this statement.

line 214: "some observations with a constant magnitude of 36": did the source paper for the observations explain the meaning of the "36"?

Section 3.1.2: please give more detail on how the 2D set was obtained from the 3D set. Also give a number for how much the computing time was reduced.

Section 3.2.2: how much longer comput time did LSTM take than SVM?

Author Response

  • line 138: "As a result, FFT does not work well for signals that continuously change and those that do not have enough samples for analysis." On the contrary FFT works well for continuous signals but not on discontinuous signals. Please correct this statement.

Reply: The statement has been corrected as advised. Updated statement in line 160 (line number changed due to added content) in section 1.3.3 to show that FFT does not work well for discontinuous signals and those that do not have enough samples for analysis.

  • line 214: "some observations with a constant magnitude of 36": did the source paper for the observations explain the meaning of the "36"?

Reply: Unfortunately, the source paper does not explain the magnitude of 36 phenomena.

  • Section 3.1.2: please give more detail on how the 2D set was obtained from the 3D set. Also give a number for how much the computing time was reduced.

Reply: Details added in section 3.1.2 on how 2D set was obtained from 3D set. Using the 3D set, the averages of the columns, which consist of the second dimension of the matrix, is taken such that an m by n by o sized-matrix becomes an m by o sized array.

It is explained in section 3.1.2 that the algorithm runs about 86% quicker as it only takes 7 seconds to train the network.

  • Section 3.2.2: how much longer compute time did LSTM take than SVM?

Reply: In section 3.2.2 it is described that training the LSTM was completed in 3.5 minutes which is 30 times longer than SVM mean features.

Reviewer 2 Report

The research in this paper is relatively new and promising.  The data  used in this paper is for real LEO space object and shows the detail realizaton of the classification using  SVM, LSTM  methods.

However, the method and conclusion is relative simple. For example, is it possible to give more analysis of the three figure 2-4, the shape, rotating condition, two relative direction from space object to the Sun and to the observer.  Are these typical for each kind of a normal satellite, tumbling satellite, and a rocket?  Why the light curves in the Figure 2 and 4 are decreasing? Is it true for every rocket body and controled satellite at any attitude? 

For Ref.22, which journal or conference is it published? 

About the classification results, is it possible to plot some 2D or 3D embedding for Chebychev or Euclidean distance  to see the separation of RSO? 

OK.

Author Response

  • The method and conclusion is relative simple. For example, is it possible to give more analysis of the three figure 2-4, the shape, rotating condition, two relative direction from space object to the Sun and to the observer. Are these typical for each kind of a normal satellite, tumbling satellite, and a rocket? Why the light curves in the Figure 2 and 4 are decreasing? Is it true for every rocket body and controlled satellite at any attitude?

Reply: Additional details have been added on the analysis of figures 2-4. In stable satellites, we will see some features of the RSO but due to the slow rotation, we may not see all faces per observation. Even so, we are able to assess when highly reflective faces, such as antennas and solar panels, are in view of the observer which results in peaks or glints. For tumbling satellites, we can see multiple peaks due to the RSO completing multiple rotations per observations. As for rocket bodies, which can be visualized as cylinders with conical tips, most of the faces have similar reflectivity so there are no distinct and wide peaks, and the brightness constantly drops in an almost linear manner. By observing the light curve more closely, it appears that there are multiple local peaks despite the overall brightness decrease during the observation. This behavior is common amongst all rocket body light curves. The similarity in the brightness pattern could be attributed the fact that the shape of the RSO is similar on multiple sides and there is no face that is significantly more reflective as opposed to solar panels and antennas on satellites. As for phase angle, it changes much more rapidly in tumbling satellite and rocket body light curves as opposed to stable satellites.

Additionally, the conclusion section has been updated to reflect the analysis above.

  • For Ref.22 which journal or conference is it published?

Reply: The paper was published in the proceedings of the 8th European Conference on Space Debris, 2021. The reference has been updated. There was an issue with all conference proceedings where the name of the conference does not appear. This issue has been resolved.

  • About the classification results, is it possible to plot some 2D or 3D embedding for Chebychev or Euclidean distance to see the separation of RSO?

Reply: A total of four additional figures have been added in a new section (2.3 Input Data Visualization) with the 3D embedding for Chebychev and Euclidean distance for both the original light curves as well as the WST coefficients. The 3D embedding of the light curves did not show a distinct separation between the different classes of RSOs except for stable satellites using Euclidean distance. By plotting the 3D embedding of the WST coefficients, there is a clearer distinction between the different RSO classes, but the groups are not completely detached visually. These plots helped visualize the complexity of the original light curve dataset and shows the improvement in using WST all while highlighting the challenges that will be encountered in classification.

Round 2

Reviewer 2 Report

It's much better than previous version. The new details show the work in this paper is rigorous and convincing, especially from Fig. 5-8.